# The Influence of Soil Physical and Chemical Properties on Saffron (*Crocus sativus* L.) Growth, Yield and Quality

Loriana Cardone [1],*, Donato Castronuovo [1], Michele Perniola [2], Laura Scrano [2], Nunzia Cicco [3] and Vincenzo Candido [1],*

[1]   School of Agricultural, Forest, Food and Environmental Sciences, University of Basilicata, Viale dell'Ateneo Lucano, 10, 85100 Potenza, Italy; donato.castronuovo@unibas.it

[2]   Department of European and Mediterranean Cultures, Environmental, and Cultural Heritage, University of Basilicata, Via Lanera, 20, 75100 Matera, Italy; michele.perniola@unibas.it (M.P.); laura.scrano@unibas.it (L.S.)

[3]   Institute of Methodologies for Environmental Analysis, National Research Council, C.da S. Loja, 85050 Tito Scalo (PZ), Italy; nunzia.cicco@imaa.cnr.it

*   Correspondence: loriana.cardone@unibas.it (L.C.); vincenzo.candido@unibas.it (V.C.)

**Abstract:** Soil physical and chemical properties play a central role in plant growth, influencing the availability of air, nutrients, and water. The aim of this two-year study was to evaluate the effect of soil texture and chemical properties (pH, electrical conductivity, organic carbon, organic matter, total, and active lime) on saffron (*Crocus sativus* L.) growth, yield, and quality. Corms were planted in pots filled with seven different soil textures obtained mixing an increasing quantity (33% and 66%) of sand to a clay soil (S1) and to a clay loam soil (S2) compared to a full (100%) sandy soil as a control (S7). A randomized complete block design comprising of seven pots with different types of soil (S1, S2, S3, S4, S5, S6, and S7) replicated three times was used. The results showed that the highest flower number (320.3 n m$^{-2}$), stigma yield (2.0 g m$^{-2}$), daughter corm production (7.9 kg m$^{-2}$), and horizontal diameter (3.1 cm) were derived from S3 and S4 soils. These were characterized by a loam and sandy-loam texture, not very calcareous, with a sub-alkaline and neutral pH, low electrical conductivity, a content of organic matter between 5.46 and 8.67 g kg$^{-1}$, and a content of active lime between 21.25 and 26.25 g kg$^{-1}$. According to International Organization for Standardization (ISO) references, although all spice samples belonged to the first qualitative category, S1, S3, and S2 soils recorded the highest value for coloring power (290.5, 289.1, and 287.6 A$^{1\%}_{1cm}$ 440 nm, respectively). The highest values of bittering (109.2 A$^{1\%}_{1cm}$ 257 nm) and aromatic (26.6 A$^{1\%}_{1cm}$ 330 nm) power were reached by S3 soil. Positive correlations were found both between color with clay and organic matter, and aroma with total calcium carbonate. In conclusion, the assessment of soil conditions is particularly important to obtain the best saffron performance in terms of stigma and daughter corms yield as well as spice qualitative traits.

**Keywords:** soil texture; organic matter; calcium carbonate; stigma yield; crocin; corm yield

## 1. Introduction

The soil fertility, i.e., capability to support plant production, is due to the interactions among physical, chemical, and biological processes. Among these, soil texture, pH, and organic matter strongly affect soil functions as well as water and nutrient availability [1,2].

The importance of soil physical quality for medicinal plant growth and yield, as well as chemical and biological conditions of the soil have been well documented in literature [3–5]. With respect to the chemical properties, nitrogen and calcium carbonate are two of the most important nutrients for crop

production and play critical roles in cell structures and plant metabolism, influencing both the content and quality of plant secondary metabolites [4,6].

The soil microbial biomass, constituted by bacteria (83%), actinomycetes (13%), fungi (3%), protozoa, algae and viruses (0.2–0.8%), improves soil structure and plant growth through the organic matter decomposition, atmospheric nitrogen fixation, nutrient cycling (C and N), and symbiosis with plants [2,7].

In saffron (*Crocus sativus* L.), a geophyte plant which propagates only via corms, the soil conditions are fundamental to allow the best stigma and daughter corm yield, and high qualitative traits of spice. This latter consists of the dried red stigmas of flower and it is used mainly in food, pharmaceutics, cosmetics, perfumery and textile industries [8].

Saffron is largely cultivated in Iran, Afghanistan, India, Greece, Morocco, Spain, Azerbaijan, and Italy where it grows on a wide range of pedologic conditions [9]. In more detail, in Iran, saffron is cultivated on calcareous soil with sandy and silty loam texture, and aridisols under irrigation conditions [10]; in Afghanistan on sandy-loam soil and rich in calcium [11]; in Azerbaijan on loamy sandy soil with low amount of organic matter [10]; in India on alfisols alkaline and silty clay loam soil with electrical conductivity from 0.09 to 0.30 dS m$^{-1}$, pH between 6.3 and 8.3, average organic carbon of 0.35%, and calcium carbonate content of 4.61% [12]; in Greece on soil from sandy clay loam to clay loam texture with pH of 7.4 and high content of calcium carbonate [13]; in Morocco on shallow, sandy-silt or calcareous clay soil with a relatively loose texture [14]; in Spain on deep, lightly calcareous and friable soil [15]; in Italy on clay loam soil or on sandy soil, with a good active calcium carbonate content, high organic substances, low phosphates and optimal potassium (Abruzzo region), on sandy or sandy-silty soil (Tuscany and Valle d'Aosta regions) [16] and, on well drained and fertile soil characterized by alluvial deposits and uniform sandy clay texture (Sardinia region) [17]. El Aymani et al. [18] report that the rhizosphere of saffron production sites in Morocco (Taliouine region) is constituted by arbuscular mycorrhizal fungi belonging to *Glomus*, *Acaulospora*, *Scutellospora*, *Gigaspora*, *Pacispora*, and *Entrophospora* genera. Parray et al. [19] show that saffron soil rhizospere in India (Kasmir region) is characterized mainly by Gram-negative bacteria as *Pseudomonas* spp., *Klebsiella* spp., *Acinetobacter haemolyticus*, *Acinetobacter lwoffii*, and *Pantoca* spp., and less by Gram-positive bacteria as *Bacillus subtilis*.

Different soil physicochemical requirements aimed to achieve the best saffron yield has been reported in literature [8,20–22]. Some authors suggest sandy to sandy loam soils [23,24], and others recommend a soil with moderate structure, clay, siliceous, ferrugenous and gipsyferous, well infiltration, with high amount of organic matter, and rich in calcium carbonate, responsible to facilitate the availability of trace elements [25,26].

Regarding to the soil pH, McGimpsey et al. [27] indicate that sixteen saffron growing sites around the world are characterized by a pH between 6.0 and 7.8, and even, that saffron soils in New Zealand has a pH from 5.2 to 5.6. While, Dhar [24] and Gresta et al. [22] report that the optimum pH is in the range from 6.8 to 7.8 (from neutral to slightly alkaline). Therefore, highly acidic and alkaline soils, and with high content of moisture are not suitable for saffron cultivation because flooding conditions could cause the corm decomposition and so a decreasing of yield [22].

A few studies about the effect of soil texture on stigma and daughter corm yield have been conducted [9]. Gresta et al. [28] evaluate three different soil textures (sand, intermediate and clay) and two corm densities (33 and 55 corm m$^{-2}$). The results show that the highest number of flowers m$^{-2}$ and stigma yield are found when corms are planted in sandy soil with high density, while the highest stigma weight is obtained in clay soil with high density. Turhan et al. [29] and Aghhavani Shajari et al. [30] report that the interaction between the growing media and fertilization influences the stigma and corm production. These authors point out that the best performances are obtained when a mixture of sand + field soil + cow manure is used for corm planting.

In addition to the stigma and corm yield, the spice quality represents an important parameter that contributes to increase the saffron economic value. Quality is determined chemically by three

main secondary metabolites: crocin ($C_{44}H_{64}O_{24}$, water-soluble crocetin esters), picrocrocin ($C_{16}H_{26}O_7$, monoterpene glycoside, a precursor of safranal), and safranal ($C_{10}H_{14}O$, a major component of essential oil), which are responsible for the color, bitter taste, and odor, respectively [31]. Currently, the method recommended by the ISO 3632 for saffron characterization, is UV–Vis spectrophotometry, which classifies the saffron in three qualitative categories [32].

Although the qualitative aspect plays a crucial role, there is no information available in literature regarding the influence of soil properties on the saffron quality, except for the study conducted by Lage and Cantrell [33]. They report that quality has not been affected by the soil chemical properties of eleven different experimental sites in Morocco, indicating only a significant positive correlation between safranal and clay content in soil.

Taking in mind the not exhaustive and not concordant data from the literature, in our opinion some aspects need to be further deepen. Thus, the present study aimed to investigate the influence of soil physical–chemical properties on the morphological traits, flowering period, stigma yield, quality traits (coloring, bittering and aromatic powers), leaf development, and daughter corms production in *Crocus sativus* L.

## 2. Materials and Methods

### 2.1. Experimental Design and Agronomic Trial

The trial was carried out for two consecutive growing seasons (2017–2018 and 2018–2019) in Potenza, located in Basilicata region Southern Italy (40°38′ N, 15°48′ E; 800 m a.s.l.), characterized by Mediterranean climate, with warm summers and cold winters (Csb/Cfb in the Köppen climate classification).

On 10 September, of each year, corms with a horizontal diameter of 3.0–3.5 cm, weighing about 15–18 g each, and from Sardinian (Italy) origin, were planted in pots filled with seven different soil textures obtained mixing an increasing quantity (33% and 66%) of sand to a clay soil (S1) and to a clay loam soil (S2) compared to a full (100%) sandy soil as a control (S7). A randomized complete block design comprising of seven pots with different types of soil (S1, S2, S3, S4, S5, S6, and S7) replicated three times was used. In detail, S1: 100% clay soil, S2: 100% clay loam soil, S3: S1 + 33% sand, S4: S2 + 33% sand, S5: S1 + 66% sand, S6: S2 + 66% sand, and S7:100% sand.

Prior to saffron planting, corms were dipped in a 1% fungicide water solution of copper oxychloride (Sumitomo Chemical Italia S.r.l., Milano, Italy) to minimize fungal diseases caused mainly by *Fusarium oxisporum* f. sp. *gladioli*. Thirty-five corms were planted in each pot with following dimensions: 37.5 cm of base diameter, 45.0 cm of top diameter, 40.0 cm of depth, and capacity of 55 L.

Weed control was carried out by hand in the autumn and spring and, the corms (including control) were irrigated (irrigation water: pH 7.2, EC 643 $\mu S$ cm$^{-1}$) (5 L per pot) during the vegetative period.

Basic fertilization was realized using a fertilizer N:P:K (20:20:20) prior to planting, while during the vegetative period, the corms were fertilized by fertigation with Hoagland nutrient solution (EC 2.5 dS m$^{-1}$; pH 6.0) two times at seven-days intervals. The solution contained the following nutrients as mmol L$^{-1}$: $NO_3^-$ 13.5; $NH_4$ 1.5; $PO_4^{3-}$ 1.0; $K^+$ 6.0; $Ca^{2+}$ 5; $Mg^{2+}$ 2.0; $SO_4^{2-}$ 2.0.

During the flowering period of each year, between October and November, flowers were collected manually, in the early hours of each day. Immediately after each harvest, flowers were taken to the 'Vegetable crops and Floriculture laboratory' of the University of Basilicata, where stigmas were separated manually and dried in a forced air oven at low temperature (40 ± 3 °C for 24 h). Dry stigmas were stored at room temperature (18 ± 2 °C) in closed glass jars and kept in the dark until qualitative analysis was performed.

The following parameters were recorded on 10 plants per plot: stigma and stamen length (mm); stigma, stamen, tepal fresh and dry weight (g); number of flowers harvested per day (n m$^{-2}$), stigma yield (g m$^{-2}$), leaf number (n plant$^{-1}$), length (cm), fresh and dry weight (g), and area (cm plant$^{-1}$). Leaf area was determined by using an area meter LI–Cor Model 3100 (LI–Cor, Inc.,

Lincoln, NE, USA), while leaf dry weight was obtained by drying samples to constant weight in a ventilated oven set at 65 °C.

At the end of each crop cycle (June 2018 and 2019), during senescence phase, the corms were lifted from the soil, cleaned and de-tunicated and then, the number of daughter corm per mother corm, fresh weight (g) and horizontal diameter of daughter corms (cm), replacement corm yield (kg m$^{-2}$) and the date of senescence were recorded. For each crop cycle, meteorological data were collected by using a weather station equipped with temperature and relative humidity probes (CS500–L– modified version of Vaisala's 50Y Humitter, Campbell Scientific Inc, Logan, UT, USA) and with a TE525 precipitation sensor (Texas Electronics, Dallas, TX, USA) to measure the rainfall. Collected data were recorded by a CR 10x data–logger (Campbell Scientific Inc, Logan, UT, USA) and were elaborated to have monthly rainfall, mean, maximum and minimum air temperature during crop cycle.

The climate parameters (maximum, mean and minimum air temperature and total rainfall) recorded in 2017–2018 and 2018–2019, from September to May, when the saffron growth occurred, are reported in Figure 1.

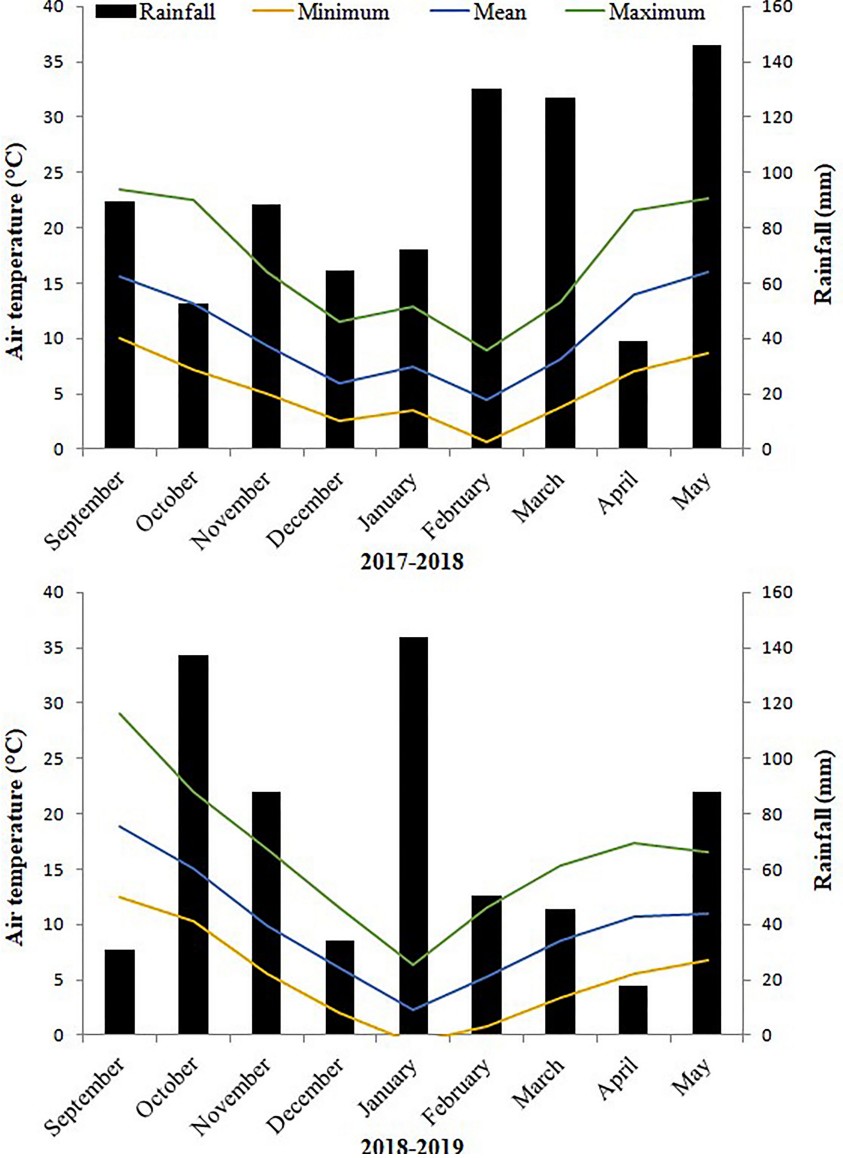

**Figure 1.** Monthly minimum, mean and maximum air temperatures and total rainfall recorded during the two growing seasons (2017–2018 and 2018–2019) in Potenza.

## 2.2. Physical-Chemical Analysis of Soil

The soil samples collected from each treatment (S1–S7) were air-dried, homogenized, passed through 2 mm sieve and analyzed for physical and chemical properties. The parameters analyzed included: (a) particle size distribution (clay, silt and sand expressed in %) and classification carried out using the USDA classification system [34] (p. 125); (b) pH measured at a soil/water ratio of 1:2.5 using a pH meter; (c) electrical conductivity measured by the conductometric method; (d) organic carbon determined by the procedure of Walkley and Black [35] and the soil organic matter content determined from the organic carbon [36]; (e) total and active lime (expressed in g $CaCO_3$ $kg^{-1}$ of soil) measured by applying the gas–volumetric method. All soil analyses were performed according to the Italian regulation [37] and the results are reported in Table 1.

**Table 1.** Physical–chemical properties of the different types of soils.

| Soil Type | Sand (%) | Silt (%) | Clay (%) | pH ($H_2O$) | EC [1] ($\mu S$ $cm^{-1}$) | Organic Carbon [2] | Organic Matter (g $kg^{-1}$) | Total Lime (g $kg^{-1}$) | Active Lime (g $kg^{-1}$) | USDA Classification | Bulk Density (g $kg^{-1}$) |
|---|---|---|---|---|---|---|---|---|---|---|---|
| S1 | 29.44 | 30.24 | 40.32 | 7.46 | 186.0 | 7.32 | 12.63 | 13.38 | 13.75 | Clay | 1.26 |
| S2 | 33.22 | 32.19 | 34.59 | 6.41 | 135.5 | 5.54 | 9.56 | 13.39 | 17.50 | Clay–loam | 1.28 |
| S3 | 52.56 | 27.85 | 19.59 | 7.64 | 272.0 | 5.03 | 8.67 | 85.86 | 26.25 | Loam | 1.38 |
| S4 | 60.31 | 21.36 | 18.33 | 7.30 | 333.0 | 3.17 | 5.46 | 85.84 | 21.25 | Sandy–loam | 1.41 |
| S5 | 69.26 | 14.05 | 16.69 | 8.11 | 169.2 | 1.18 | 2.04 | 145.3 | 23.75 | Sandy–loam | 1.45 |
| S6 | 81.67 | 9.32 | 9.01 | 7.61 | 254.0 | 1.19 | 2.05 | 92.43 | 26.25 | Loamy–sand | 1.51 |
| S7 | 90.61 | 2.27 | 7.12 | 8.10 | 86.1 | 0.00 | 0.00 | 123.33 | 26.75 | Sandy | 1.55 |

[1] Electrical conductivity; [2] Walkley–Black method.

## 2.3. Spectrophotometric Analysis of Saffron Extract

Samples of spice produced in 2017 and 2018 were analyzed in triplicate according to the ISO 3632 normative [32]. In detail, 500 mg of powdered samples, previously passed through a 0.5 mm sieve, were transferred into a 1000 mL volumetric flask and 900 mL of distilled water were added to obtain an aqueous saffron extract. The aqueous solution was stirred for 1 h in the dark and then brought to 1000 mL with distilled water. The obtained saffron extract was diluted 1:10 with deionized water and filtrated with polytetrafluoroethylene (PTFE) filters (15 mm diameter and 0.45 μm pore size). Successively, the aqueous saffron extract was analyzed by using an UV-Vis spectrophotometer. Picrocrocin, safranal, and crocin were obtained by direct readings of the specific absorbance of 1% aqueous saffron extract at 257, 330, and 440 nm, respectively, using the following equation:

$$A^{1\%}_{1cm} = \frac{(D \times 20,000)}{(100 - H)}$$

(1)

where $D$ is the absorbance at 257, 330, and 440 nm; 20,000 is the total extract dilution considering the amount of saffron sample; $H$ is the moisture and volatile matter content, expressed as a mass fraction. According to the ISO 3632 [32], $H$ was determined placing 2.5 ± 0.001 g of each saffron samples in an oven from 103 ± 2 °C for 16 h and it was calculated as the percentage of the initial weight of the sample according to the following formula:

$$H = (m_0 - m_1) * (100/m_0) \%$$

(2)

where $m_0$ is the mass, in grams, of the saffron portion before drying and $m_1$ is the mass, in grams, of the dry residue.

## 2.4. Color Measurement of Dry Stigma and Fresh Tepal

Color of dry stigma was determined according to the methodology by Cuko et al. [38] and measured by means of a Minolta CR–400 Chroma Meter (Minolta Corp., Osaka, Japan). The colorimeter was calibrated using a standard white plate. Fifteen stigma samples were used for each treatment (soil type) and five readings were made on each set of sample. Color coordinates were expressed as L*

describing lightness (L* = 0 for black, L* = 100 for white), a* describing intensity in green–red (a < 0 for green, a > 0 for red), b* describing intensity in blue–yellow (b < 0 for blue, b > 0 for yellow) in the CIELAB color system [39].

In addition, color of fresh tepal was measured on the center of the upper tepal surface according to Castronuovo et al. [40].

*2.5. Statistical Analysis*

Data were subjected to analysis of ANOVA procedure, considering 'soil type' and 'year' as sources of variation. Mean values were separated by Student Newman–Keuls (SNK) test at $p \leq 0.05$. Principal component analysis (PCA) was performed to evaluate correlation between soil properties with morphological, quantitative and qualitative traits of saffron. Statistical analyses were performed using the software RStudio: Integrated Development for R, version 1.0.136 [41].

# 3. Results

*3.1. Climatic Data*

As reported in Figure 1, the first saffron crop cycle (2017–2018), with an annual average rainfall of 809.6 mm, was more rainy than the second one (636.2 mm).

In particular, the first year was characterized by higher rainfall (433.2 mm) during the vegetative period (from December to April) than the second one (292.4 mm).

Meanwhile, in the second saffron crop cycle (2018–2019), rainfall was particularly concentrated and higher (225.2 mm) during the flowering period (from October to November) compared to the previous year (140.8 mm).

Monthly minimum, mean, and maximum values of air temperatures recorded during the two experimental years were nearly similar. In particular, the second year was characterized by lower air temperatures during the vegetative period. This meteorological trend was due mostly to the monthly mean temperature recorded in January (2.3 °C) and in April (10.7 °C) compared to those recorded in the first year (7.4 and 14.0 °C, respectively) (Figure 1).

*3.2. Morphological and Colorimetric Traits of Flower*

As shown in Table 2, types of soil had a significant effect on the major morphological traits of flower, except for some color coordinates of tepal (a* and b*). The highest values of stamen length, fresh and dry weight were obtained in S4, while the highest values of tepal fresh and dry weight were obtained in S1 followed by S2 and S4 soils.

In addition, S4 showed the highest dry weight of floral bioresidues, consisting of stamen and tepal. Concerning to colorimetric coordinates of fresh tepal, the lowest values of L* were recorded in flowers collected in S1 followed by S3 soil.

Significant differences between the two experimental years were observed. Particularly, in 2018 flowers showed higher values of some parameters such as stamen fresh weight and tepal dry weight (Table 2).

All the morphological traits, except for stamen length and colorimetric coordinates of tepal (L*, a* and b*), were influenced both by the soil type and year, as highlighted by the significance of their interaction ($p < 0.05$; $p < 0.01$) (Table 2).

*3.3. Flowering Period and Stigma Yield*

Table 3 shows that the soil type significantly influenced the flowering interval, stigma traits and yield. Although no significant effect of soil type on flowering earliness was found, the flowering occurred before in soil characterized by a low amount of sand (from S1 to S4).

**Table 2.** Effect of the soil type on morphological and colorimetric traits of saffron flower in two years.

| Treatments [1] | Stamen Length (mm) | Stamen Fresh Weight (g) | Stamen Dry Weight (g) | Tepal Fresh Weight (g) | Tepal Dry Weight (g) | By-Products Dry Weight (g) | L* | a* | b* |
|---|---|---|---|---|---|---|---|---|---|
| **Soil type (S)** | | | | | | | | | |
| S1 | 20.8 a | 0.0405 a | 0.0076 c | 0.316 a | 0.039 a | 0.0466 ab | 48.6 c | 26.2 a | −19.8 a |
| S2 | 20.6 a | 0.0360 b | 0.0042 d | 0.308 b | 0.037 ab | 0.0412 c | 51.9 ab | 25.4 a | −20.7 a |
| S3 | 19.5 b | 0.0348 c | 0.0086 b | 0.299 c | 0.034 b | 0.0426 c | 49.6 bc | 25.3 a | −19.3 a |
| S4 | 20.4 a | 0.0389 ab | 0.0107 a | 0.309 b | 0.037 ab | 0.0477 a | 51.6 ab | 25.8 a | −20.4 a |
| S5 | 19.3 b | 0.0311d | 0.0085 b | 0.300 c | 0.036 ab | 0.0445 b | 51.2 ab | 25.4 a | −19.6 a |
| S6 | 20.6 a | 0.0343 c | 0.0056 cd | 0.299 c | 0.033 b | 0.0386 d | 52.7 a | 25.4 a | −19.4 a |
| S7 | 18.5 c | 0.0238 e | 0.0072 c | 0.279 d | 0.029 c | 0.0362 d | 51.7 ab | 26.1 a | −20.2 a |
| *Significance* [2] | ** | *** | *** | *** | ** | ** | ** | NS | NS |
| **Years (Y)** | | | | | | | | | |
| 2017–2018 | 19.0 | 0.0247 | 0.0074 | 0.299 | 0.033 | 0.0405 | 51.5 | 25.1 | −19.3 |
| 2018–2019 | 21.0 | 0.0437 | 0.0076 | 0.304 | 0.038 | 0.0445 | 50.6 | 26.2 | −20.5 |
| *Significance* | NS | *** | NS | NS | ** | ** | NS | NS | NS |
| **Interaction S × Y** | | | | | | | | | |
| *Significance* | NS | ** | ** | ** | * | * | NS | NS | NS |

[1] Values followed by a different letter are significantly different at $p \leq 0.05$, according to SNK test. [2] *, significance at $p < 0.05$; **, significance at $p < 0.01$; ***, significance at $p < 0.001$; NS, no significant difference.

**Table 3.** Effects of the soil type on flowering and stigma yield in two years.

| Treatments [1] | Days to Flower (d) | Flowering Interval (d) | Stigma Length (mm) | Stigma Fresh Weight (g) | Stigma Dry Weight (g) | Flowers (n corm$^{-1}$) | Flowers (n m$^{-2}$) | Stigma Yield (g m$^{-2}$) |
|---|---|---|---|---|---|---|---|---|
| **Soil type (S)** | | | | | | | | |
| S1 | 59.3 a | 17.2 a | 37.7 a | 0.0426 a | 0.0071 a | 1.3 a | 279.2 b | 2.0 a |
| S2 | 59.7 a | 12.5 c | 36.2 ab | 0.0396 ab | 0.0066 ab | 1.2 a | 269.8 b | 1.8 ab |
| S3 | 59.7 a | 12.5 c | 35.9 ab | 0.0378 ac | 0.0063 ac | 1.4 a | 312.5 ab | 2.0 a |
| S4 | 59.7 a | 13.7 bc | 35.6 ab | 0.0378 ac | 0.0063 ac | 1.5 a | 328.1 a | 2.0 a |
| S5 | 61.2 a | 15.7 ab | 33.4 ac | 0.0354 bc | 0.0059 bc | 1.3 a | 280.2 b | 1.7 ab |
| S6 | 61.2 a | 16.5 ab | 31.9 bc | 0.0330 c | 0.0055 c | 1.2 a | 265.6 b | 1.5 b |
| S7 | 61.5 a | 14.3 ac | 28.8 c | 0.0234 d | 0.0039 d | 0.9 b | 190.6 c | 0.8 c |
| *Significance* [2] | NS | * | ** | ** | ** | ** | * | ** | ** |
| **Years (Y)** | | | | | | | | |
| 2017–2018 | 64.0 | 16.1 | 31.9 | 0.0378 | 0.0063 | 1.2 | 271.4 | 1.7 |
| 2018–2019 | 56.6 | 13.2 | 36.4 | 0.0336 | 0.0056 | 1.3 | 278.9 | 1.6 |
| *Significance* | ** | * | ** | ** | * | NS | NS | NS |
| **Interaction S × Y** | | | | | | | | |
| *Significance* | ** | ** | ** | ** | * | NS | ** | ** |

[1] Values followed by a different letter are significantly different at $p \leq 0.05$, according to SNK test. [2] *, significance at $p < 0.05$; **, significance at $p < 0.01$; ***, significance at $p < 0.001$; NS, no significant difference.

In general, flowering began after 60.3 days from planting and continued, on average, for 14.6 days. In detail, during 2017, the flowering started about nine days later than 2018 (Figure 2a). S1 bloomed before, reaching 93.6 flowers $m^{-2}$ in the first production peak on October 30. In contrast, S7 and S6 reached the first production peak after five days from S1 with 60.4 and 62.5 flowers $m^{-2}$, respectively (Figure 2a).

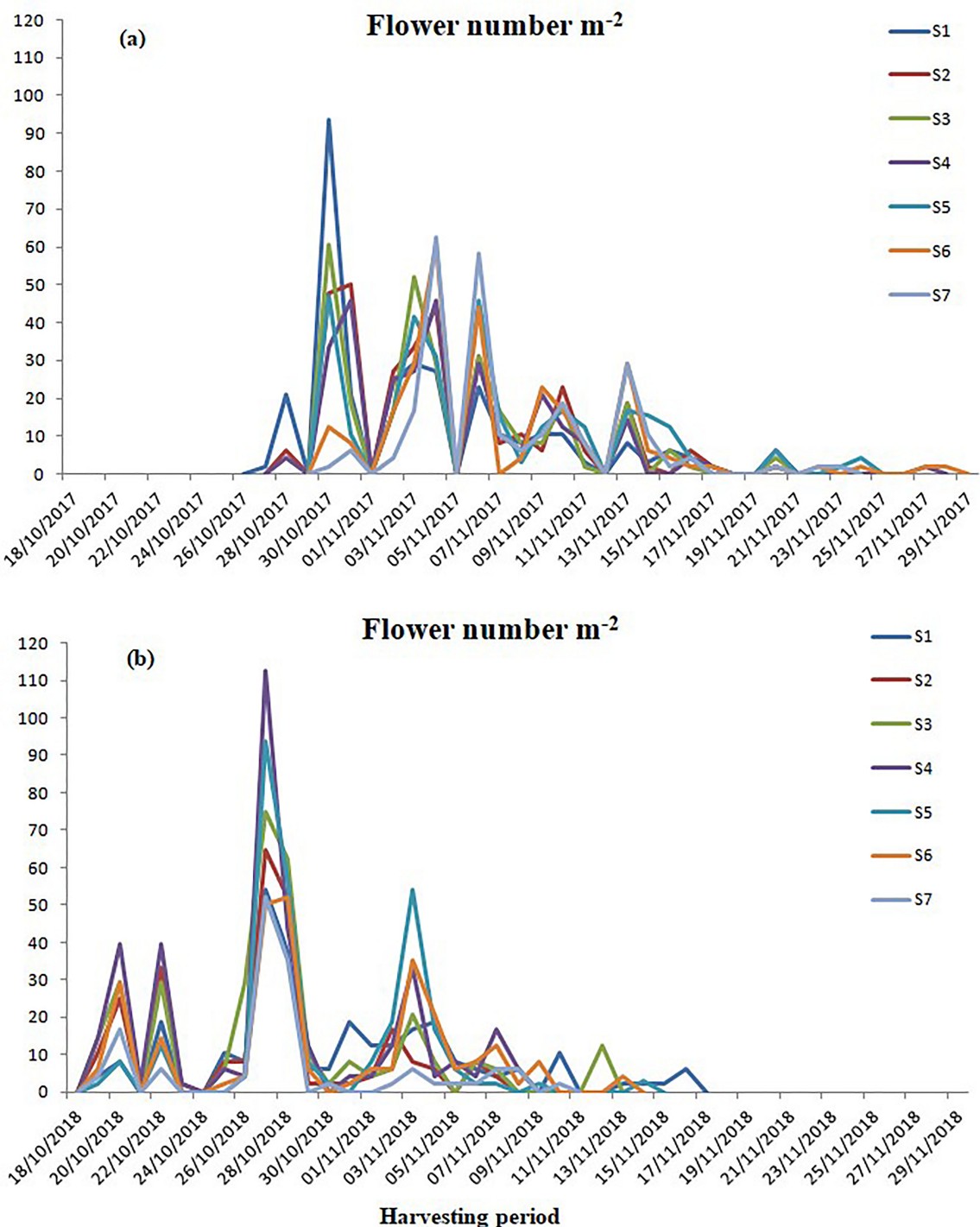

**Figure 2.** Flowering period during 2017 (**a**) and 2018 (**b**) for all type of soil.

In 2018, the flowering occurred for a shorter period and was characterized by higher number of flowers m$^{-2}$ than the flowering occurred in 2017 (Table 4). Although in the second year the flowering started in the same time for all soil types, S4 recorded the highest cumulative number of flower m$^{-2}$ (220.8) followed by S3 (183.3) in the first ten days of harvesting period (Figure 2b).

Concerning the flowering interval, a significant 'soil type × year' interaction occurred: S1 showed the widest flowering interval in 2018 (18 days) (Figure 2b and Table 3).

Stigma length, fresh and dry weight had a significant increase as the amount of clay and organic matter raised, in fact, S1 and S2 soils produced longer and heavier stigmas than S5, S6 and S7.

Types of soil had a low impact on the number of flowers per corms, even if the highest and lowest values were recorded in S4 and S7 soils, respectively. Considering the effect on the number of flower m$^{-2}$ and stigma yield, S4 reached the highest values followed by S3 (Table 3).

The dry stigma was 16.7% of fresh weight in all soil types. In general, in this study 140,845–256,410 flowers were needed, and 6.7–9.3 kg of dry by-products were achieved to obtain 1 kg of dry stigmas (Table 3).

**Table 4.** Effects of the soil type on spice qualitative and colorimetric traits in two years.

| Treatments [1] | Qualitative Traits | | | | | | Color Coordinates | | |
|---|---|---|---|---|---|---|---|---|---|
| | $A^{1\%}_{1cm}$ (440 nm) [3] | ISO Reference (Crocetin Esters) [4] | $A^{1\%}_{1cm}$ (257 nm) [5] | ISO Reference (Picrocrocin) [6] | $A^{1\%}_{1cm}$ (330 nm) [7] | ISO Reference (Safranal) [8] | L* | a* | b* |
| **Soil type (S)** | | | | | | | | | |
| S1 | 290.5 a | I | 107.9 a | I | 23.6 c | I | 36.5 c | 38.6 a | 36.9 c |
| S2 | 287.6 a | I | 105.4 a | I | 24.1 ac | I | 36.9 c | 36.8 c | 37.5 c |
| S3 | 289.1 a | I | 109.2 a | I | 26.6 a | I | 37.7 bc | 37.2 bc | 38.5 bc |
| S4 | 275.6 b | I | 106.4 a | I | 25.4 ab | I | 37.7 bc | 38.2 ab | 38.5 bc |
| S5 | 271.7 b | I | 105.9 a | I | 24.8 b | I | 38.1 bc | 37.3 bc | 39.8 ab |
| S6 | 250.6 c | I | 96.2 b | I | 24.7 b | I | 38.9 b | 37.8 ac | 40.2 ab |
| S7 | 247.4 c | I | 95.1 b | I | 24.9 b | I | 40.9 a | 38.9 a | 41.7 a |
| *Significance* [2] | ** | | * | | * | | ** | ** | ** |
| **Years (Y)** | | | | | | | | | |
| 2017–2018 | 266.3 | I | 103.6 | I | 21.2 | I | 40.2 | 38.8 | 41.0 |
| 2018–2019 | 280.1 | I | 103.9 | I | 28.6 | I | 36.5 | 36.8 | 38.0 |
| *Significance* | ** | | NS | | ** | | * | NS | NS |
| **Interaction S × Y** | | | | | | | | | |
| *Significance* | * | | * | | * | | * | NS | NS |

[1] Values followed by a different letter are significantly different at $p \leq 0.05$, according to SNK test. [2]*, significance at $p < 0.05$; **, significance at $p < 0.01$; ***, significance at $p < 0.001$. NS, no significant difference. [3] Absorbance of 1% aqueous saffron extract at 440 nm. [4] ISO reference for crocetin esters: I category $A^{1\%}_{1cm} \geq 200$, II category $A^{1\%}_{1cm} \geq 170$, III category $A^{1\%}_{1cm} \geq 120$. [5] Absorbance of 1% aqueous saffron extract at 257 nm. [6] ISO reference for picrocrocin: I category $A^{1\%}_{1cm} \geq 70$, II category $A^{1\%}_{1cm} \geq 55$, III category $A^{1\%}_{1cm} \geq 40$. [7] Absorbance of 1% aqueous saffron extract at 330 nm. [8] ISO reference for safranal: I, II and III category $A^{1\%}_{1cm}$ minimum 20 and maximum 50.

### 3.4. Qualitative and Colorimetric Traits of Spice

As shown in Table 4, qualitative traits were significantly affected by types of soil. In general, all samples belonged to the first category according to ISO 3632 references [32]. The highest values of coloring power were reached by S1, S3, and S2 soils, while the highest bittering and aromatic powers were achieved by S3 soil. In contrast, the lowest values of color and taste were obtained by S6 and S7 soils, meanwhile the decrease of aroma was found in saffron from S1 and S2 soils.

Concerning the colorimetric traits of dry stigma, all coordinates were significantly influenced by soil types (Table 4). In particular, S6 and S7 showed greater values of lightness (L*) and yellowness (b*).

Experimental year influenced coordinate L*, coloring and aromatic powers ($p < 0.05$). The spice obtained during the second crop cycle showed a significant increase in crocin ($A^{1\%}_{1cm}$ 440 nm) and safranal ($A^{1\%}_{1cm}$ 330 nm), and a decrease in lightness (Table 4).

### 3.5. Leaf Traits and Daughter Corm Yield

The results showed that types of soil influenced significantly all the leaf traits (Table 5). Among these, leaf length, fresh and dry weight, number, and area plant$^{-1}$ were all the highest ones in S1, followed by S2 and S3 soils. Some traits, such as leaf length, number and area were influenced by experimental year ($p < 0.001$). The results highlighted that leaf length and area decreased by 14% and 12%, respectively, while the leaf number increased by 34% during the second year (Table 5).

**Table 5.** Effects of the soil type on leaf traits of saffron in two years.

| Treatments [1] | Leaf Length (cm) | Leaf Fresh Weight (g) | Leaf Dry Weight (g) | Leaf (n plant$^{-1}$) | Leaf Area (cm$^2$ plant$^{-1}$) |
|---|---|---|---|---|---|
| **Soil type (S)** | | | | | |
| S1 | 49.8 a | 0.557 a | 0.152 a | 27.6 a | 192.4 a |
| S2 | 43.4 b | 0.521 bc | 0.137 bc | 22.9 ac | 180.8 b |
| S3 | 43.9 b | 0.499 c | 0.131 c | 26.1 ab | 140.3 d |
| S4 | 42.6 b | 0.542 ab | 0.143 ab | 23.8 ac | 160.8 c |
| S5 | 34.4 c | 0.319 d | 0.089 d | 20.7 c | 98.6 e |
| S6 | 34.2 c | 0.318 d | 0.081 d | 20.7 c | 107.9 e |
| S7 | 26.9 d | 0.234 e | 0.066 e | 21.1 bc | 74.4 f |
| *Significance* [2] | *** | *** | *** | * | *** |
| **Years (Y)** | | | | | |
| 2017–2018 | 42.3 | 0.434 | 0.111 | 19.9 | 145.2 |
| 2018–2019 | 36.4 | 0.421 | 0.117 | 26.7 | 127.5 |
| *Significance* | * | NS | NS | *** | ** |
| **Interaction S × Y** | | | | | |
| *Significance* | *** | *** | *** | *** | *** |

[1] Values followed by a different letter are significantly different at $p \leq 0.05$, according to SNK test. [2] *, significance at $p < 0.05$; **, significance at $p < 0.01$; ***, significance at $p < 0.001$; NS, no significant difference.

The lowest number of daughter corm per mother corm was recorded in S2 (Table 6). The greatest daughter corm mean weight, horizontal diameter and yield were obtained when saffron grew in S3 and S4 soils (Table 6). These soils increased the daughter corm yield by 40.6% compared to the control (S7). Soil 'type × year' interaction was statistically significant ($p < 0.05$; $p < 0.01$) for all the daughter corm traits. In particular, the highest daughter corm yield (8.88 kg m$^{-2}$) was reached by S4 soil during the first year (data not shown).

The growth cycle ranged between 241.0 and 252.0 days and it was influenced by types of soil and experimental year. In detail, a longer growth cycle and a later senescence were observed in S5, S6, and S7 (Table 6).

In the second experimental year, the senescence occurred about ten days before, associated by an increase of daughter corm number by 48% and a decrease of horizontal diameter by 37% compared to the first year.

**Table 6.** Effects of the soil type on daughter corm traits and yield in two years.

| Treatments [1] | Daughter Corm (n Mother Corm$^{-1}$) | Daughter Corm Mean Weight (g) | Total Daughter Corm Weight (g Plant$^{-1}$) | Daughter Corm Horizontal Diameter (cm) | Daughter Corm Yield (kg m$^{-2}$) | Growth Cycle (d) |
|---|---|---|---|---|---|---|
| **Soil type (S)** | | | | | | |
| S1 | 3.2 a | 13.5 a | 35.0 ab | 2.9 ab | 7.7 ab | 244.5 b |
| S2 | 2.5 b | 10.2 ab | 24.9 cd | 2.9 ab | 5.5 cd | 244.5 b |
| S3 | 3.4 a | 14.6 a | 35.7 ab | 3.2 a | 7.8 a | 245.0 b |
| S4 | 3.4 a | 14.4 a | 36.7 a | 3.1 a | 8.0 a | 245.0 b |
| S5 | 3.5 a | 9.7 ab | 29.1 bc | 2.7 ab | 6.4 bc | 248.5 ab |
| S6 | 3.6 a | 9.5 ab | 25.7 cd | 2.6 ab | 5.6 cd | 248.5 ab |
| S7 | 3.4 a | 6.3 b | 21.4 d | 2.3 b | 4.7 d | 251.5 a |
| *Significance* [2] | * | * | ** | * | *** | * |
| **Years (Y)** | | | | | | |
| 2017–2018 | 2.2 | 15.2 | 31.7 | 3.3 | 6.93 | 252.3 |
| 2018–2019 | 4.3 | 7.1 | 28.0 | 2.4 | 6.12 | 241.3 |
| *Significance* | *** | *** | NS | ** | NS | * |
| **Interaction S × Y** | | | | | | |
| *Significance* | ** | ** | ** | * | ** | * |

[1] Values followed by a different letter are significantly different at $p \leq 0.05$, according to SNK test. [2] *, significance at $p < 0.05$; **, significance at $p < 0.01$; ***, significance at $p < 0.001$; NS, no significant difference.

## 3.6. Principal Component Analysis

Principal component analysis (PCA) was performed to evaluate correlation between physical-chemical properties of soil (sand, silt, clay, pH, electrical conductivity, total lime, organic matter) with morphological (by-products dry weight, leaf area, number of daughter corms per plant, daughter corms weight and diameter), quantitative (spice and daughter corm yield) and qualitative (L*, color, bitter and aroma) traits. Eighteen original variables were analyzed by employing PCA, and were reduced to two principal components, which represent 86.8% of the total variability. In detail, the first component (PC 1) accounted for the 65.2% of the total variability, and the second one (PC 2) accounted for the 21.6% (Table 7).

**Table 7.** Loadings of the significant variables on two first principal components from analysis of morphological, quantitative and qualitative data.

| Variables | Principal Components | |
|---|---|---|
| | 1 | 2 |
| Sand | −0.9324 | 0.3465 |
| Silt | 0.9699 | −0.1664 |
| Clay | 0.8446 | −0.4923 |
| pH | −0.8048 | 0.2333 |
| Total lime | −0.7271 | 0.5592 |
| Organic matter | 0.9207 | −0.2561 |
| Electrical conductivity | 0.4197 | 0.7920 |
| By-products weight | 0.8051 | 0.2875 |
| Leaf area | 0.9662 | −0.1156 |
| Daughter corm number | −0.4839 | 0.7433 |
| Daughter corm weight | 0.8518 | 0.4890 |
| Daughter corm diameter | 0.8870 | 0.4031 |
| Stigma yield | 0.7673 | 0.5823 |
| Daughter corm yield | 0.7456 | 0.5973 |
| L* | −0.9122 | 0.1132 |
| Color | 0.9505 | −0.0609 |
| Taste | 0.8936 | 0.2126 |
| Aroma | −0.0719 | 0.7902 |
| Eigenvalue | 11.73 | 3.88 |
| Total variance (%) | 65.20 | 21.57 |

The first PC was defined by clay, silt, organic matter, spice yield, daughter corm diameter and two qualitative parameters (color and taste). In fact, in the loading plot (Figure 3) these variables were placed far from the origin of the first PC, to the right in the loading plot, close together and, therefore, positively correlated. The first PC was characterized by a negative relationship of stigma yield, corm horizontal diameter, and crocin with sand and pH.

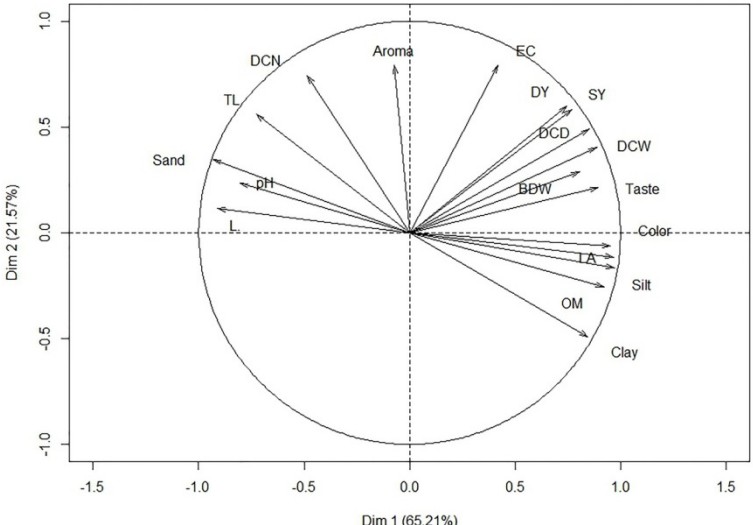

**Figure 3.** Correlation biplot of the two first principal components variables in the two-dimensional space. TL: Total lime; EC: Electrical conductivity; OM: Organic matter; DY: Daughter corm yield; SY: Stigma yield; DCN: Daughter corm number; DCW: Daughter corm weight; DCD: Daughter corm diameter; BDW: Bioresidues dry weight; LA: Leaf area.

The second PC was characterized by total lime, electrical conductivity, daughter corm number and aroma as qualitative parameter.

The score plot (Figure 4) shows a good separation between types of soil. In particular S1, S2, S3 and S4 were clearly separated from S5, S6, and S7. These latter soils tended to be differentiated from the other types of soil on the negative side of the PC1 axis and were characterized by lower value of saffron and daughter corms yield, quality (color and taste) and by higher content of sand, values of total lime, pH, and number of daughter corms per plant.

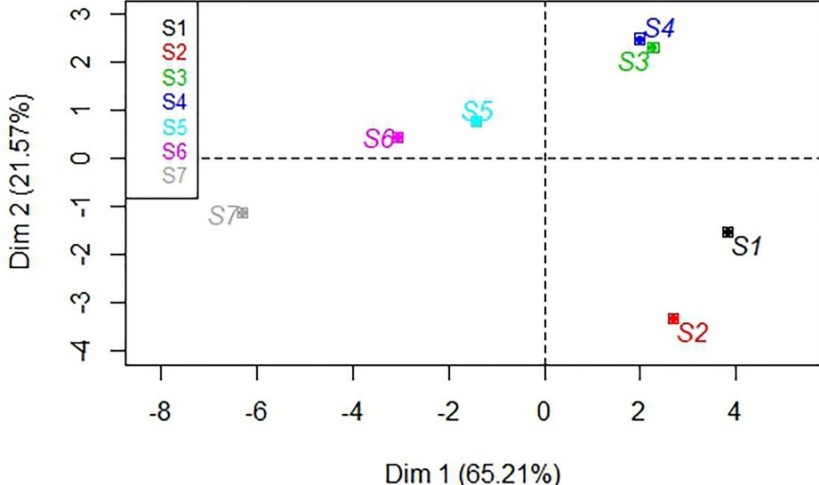

**Figure 4.** Biplot showing the projection of from the analysis of soil properties, morphological, quantitative and qualitative traits.

A positive and significant correlation between organic matter, clay, and silt with dry weight by-products, leaf area, spice and corm yield, diameter and weight of daughter corms, color and taste was found.

The correlation analysis (Figure 3) revealed that the coloring power displayed a negative correlation with colorimetric coordinate L*, indicating that the spice obtained in S6 and S7 was clearer.

The trend of aromatic power was different from the other powers: it was positively correlated with total calcium carbonate, pH, and sand, but negatively correlated with clay and organic matter (Figure 3).

## 4. Discussion

### 4.1. Flowering and Quantitative Traits

The present study was conducted to evaluate the effect of soil texture and chemical properties on the growth, yield and quality of saffron.

Concerning the morphological traits, S1, characterized by a clay texture, slightly alkaline, with an organic matter of 12.63 g kg$^{-1}$ and not very calcareous, obtained darker tepals of flower. Soils with this texture are colder and low temperatures could to have increased the accumulation of anthocyanins and so the color in tepals [42].

Although S1 showed the highest values of flower and leaf traits, especially due to its greater content of organic matter, it did not show the best results in terms of daughter corm production. In accord with Gresta et al. [28], we consider that the soil heavy texture could reduce the root growth and so the saffron productivity. Indeed, as reported by Chaudhari et al. [43], the bulk density, negatively correlated with organic matter, total macronutrient, and micronutrient contents, causes a root restriction for clay soil at value of 1.4 g cm$^{-3}$.

Our results were influenced by the different climatic conditions of two experimental years. In particular, the second crop cycle was characterized by an early and shorter flowering and, by a production of flower with longer stigma and heavier stamen and tepal. In accordance with Gresta et al. [22] and as reported in our previous study [44], flowering time and flower traits were positively influenced by the higher rainfall recorded during the harvesting period in 2018.

In addition, the early flowering, occurred in soils with higher content of clay and silt, could be due to lower temperatures and higher soil moisture compared to soils richer in sand [28].

Soil properties influenced partially the flower number per corm because the corm is the primary nutrient reserve for stigma production. The contribution of corms to produce flowers depends especially on their dimension and corm geographical origins [44,45].

Saffron yield is a parameter which depends on many factors as agronomic aspects (density planting, fertilization, irrigation, weed control) [46,47], climatic aspects (air temperature and precipitations), and cultivation site conditions [8,44]. Among soil properties, the texture is the major factor stimulating saffron production [48], because controls the movement and availability of air, nutrients and water [49].

PCA analysis showed that S4 and S3 soils had the best performance in terms of stigma and daughter corms yield. These soils, characterized by sandy loam and loam texture, respectively, with neutral-sub alkaline pH and a good amount of organic matter, present positive aspects as an average water-holding capacity, good permeability, fair resistance to drought, and ease of working. In contrast, S7, S6, and S5 highlighted lower values of yields. These soils have a higher content of sand, where porosity is favorable, but chemical properties as nutrient supplying capacity is low.

The results obtained from this study are in agreement with Liu et al. [50] who reported that loam soil was the suitable soil type for the cultivation of root/stem- types of medicinal plants. Similar results were also observed by Khorramdel et al. [51], reporting that stigma yield in sandy loam soil was higher than in clay loam by 49% under greenhouse experiments in Iran.

Stigma yield depends on the soil fertility (16–80%) as the organic matter content, which plays an important role in maintaining physical, chemical, and biological properties of soil [52]. In according to

Koocheki et al. [47], organic matter positively affected the leaf development, stigma, and daughter corm yield.

This study shows that saffron tolerated a wide pH range of soil, from slightly acid to alkaline. Therefore, this result is in agreement with McGimpsey et al. [27] as well as with our previous work [38], where the highest stigma yield was obtained in soils with acidic and slightly alkaline pH, respectively.

The evaluation of soil conditions in the context of daughter corm yield results more important since saffron is propagated only vegetatively. The growth of daughter corms depends on mother corm size [53], photosynthetic capacity of leaves (87–91%), mother corm reserves (9–13%), cultivation conditions and water soil content [45]. The higher water availability during daughter corms formation in the first year allowed to increase the leaf length, area and photosynthetic rate, and to produce daughter corms with higher diameter and weight [45,54].

The S3 and S4 textures, above mentioned, favored the growth and the multiplication of daughter corms, reaching a diameter able to produce flowers. Some studies have shown similar results in other species, as in turmeric [55] and in potato [56].

Regarding the soil biological properties, microbial biomass and activity are influenced by soil properties, as texture, structure, porosity, pH, moisture, organic matter content and available phosphorus, and by management practices as type and age of plantation and fertilization [57–61]. In S3 and S4 soils, the neutral-sub alkaline pH could have favored the bacterial growth and, the content of clay and silt could have protected the microbial biomass from predation, desiccation, heat and fluctuations of water availability and allowed soil enzyme activities (urease, invertase and alkaline phosphatase) [62–64]. Among the bacteria, *B. subtilis* could promote and improve stigma yield and corm growth through phytohormones (i.e., indoleacetic acid) and phosphate solubilization production [19]. Other important components of the soil are fungi. *Funneliformis mosseae*, showing preferences in higher pH soil (6.6–7.4) [65], could have contributed to increasing flower production and stigma yield in S3 and S4 soils [66].

Our quantitative data are higher than those reported in other studies because of greater planting density used. Indeed, these values of stigma and daughter corm yield were generally reached in a saffron pluriannual growing cycle, especially in a two-year [44] or using large corms (38–53 g) [53].

The early plant senescence in soils characterized by a high content of clay (S1–S2) could be explained by the increased ethylene production, which is caused by a possible root restriction [67]. In addition, the early senescence occurring in the second year could be due to climatic conditions, that is the lower rainfall recorded during the growing period than the first year.

With respect to the effect of soil type on the by-products dry weight, it can be observed that S4 obtained the highest value, especially due to the greater length and weight of stamen. The combination between clay and organic matter content of S4 soil was been the most suitable among from those investigated, favoring the formation of flower, and increasing its morphological traits [5].

Floral bio-residues could contribute to supplementing the income of growers by using them in the cosmetic and pharmaceutical industries thanks to antioxidant, anti-inflammatory, and antidepressant activity [68,69]. Indeed, besides spice and corms market, another approach that can increase the profitability of saffron production is the valorization and use of its by-products, respecting the circular economy principles [9].

### 4.2. Qualitative Traits

Being a medicinal plant, it is not only important to achieve a greater yield, but also to obtain mainly a constant active compound content in order to have a better and uniform stigma quality [4,31].

Many studies on the effect of environmental conditions [33,44], harvesting time [70], "mondatura", drying [71], and storage methods [72] on saffron quality have been conducted but no significant influence of soil chemical properties has been reported.

The variation in crocin, picrocrocin and safranal has been demonstrated to be associated with the soil characteristics. This result, in agreement to what reported in our previous study [44],

further highlighted the significant effect of cultivation site conditions on saffron quality, in particular on coloring and bittering powers. In addition, the decrease of crocin and picrocrocin values obtained on sandy and loamy-sand soils could be justified by more loose structure of these soils, giving lower resistance to root growth and thereby reducing the production of stress-related color and bitter compounds [73].

A decrease of color and taste and increase of aroma were found in alkaline, medium calcareous, and poor in organic matter soil. This finding was supported by the study done by Arnò et al. [74], who evaluated the effect of soil and crop nutrition characteristics on grape yield and quality. They found that soil with an excessive content of calcium carbonate demonstrated a limitation in terms of the availability of microelements, such as manganese, iron, and boron, which caused a reduction in phenolic content and, subsequently, in color of grapes.

The main compound affecting saffron aroma is the safranal, a cyclical terpenic aldehyde, produced through enzymatic and thermal degradation from picrocrocin during the drying and storage phases [75]. Some authors reported the significant effect of soil properties on the variation in essential oil composition of different officinal plants [4,76–78] and a positive correlation between volatile terpenes and calcium carbonate was found also in other crops. Moretti et al. [79] found that higher content of 1,8-cineole essential oil in *Rosmarinus officinalis* L., grown in very calcareous soil, was an adaption mechanism to unsuitable conditions to crop growth, as nutrient deficiencies. Ormeno and Fernandez [80], reported that *Pinus halepensis* Mill., *Cistus halbidus* L., and *Myrtus communis* L. showed higher terpenes content when growing in calcareous soil compared to siliceous one. Mumivand et al. [6] showed that calcium carbonate increased the concentration of major components of essential oil (carvacrol, y-terpinene, and β-bisabollene) in *Satureja ortensis* L.

Organic matter was found to improve the biosynthesis of secondary metabolites in many medicinal and aromatic plants [81,82]. Saffron showed high coloring and bittering powers when growing in soils that contained a good organic matter because increasing nutrient availability of basic elements (i.e., nitrogen) and/or to that of cation exchange capacity (CEC) could favor the crocin biosynthesis and its accumulation in stigmas [4]. This finding is in agreement with that obtained by Rezaian and Paseban [83], who reported that crocin and picrocrocin content increased by the application of cow manure while the safranal concentration decreased compared to the control. Rabani-Foroutagheh et al. [84] indicated that foliar fertilization with 'Agrimel', characterized by higher nitrogen and phosphorus contents, enhanced the saffron quality showing a positive effect on crocin content but a negative effect on safranal.

Coloring strength is one of the most important qualitative parameters for consumer and for market price [85] and it can be evaluated also through a rapid determination by tristimulus CIELAB [86]. A negative relationship between the coordinate L* and coloring power was found also in tomato crop. Arias et al. [87] showed that the decrease of L* with maturity reflected the darkening of the tomatoes with carotenoid synthesis and the loss of greenness.

Regarding the climatic effect, the higher air temperature recorded during flowering in the second experimental year favored the crocin biosynthesis and the improvement of coloring power of spice [44].

## 5. Conclusions

Saffron is commercially cultivated for medicinal and culinary uses of dry stigma, and for the daughter corms as the only propagation material. Although the saffron tolerates a wide range of pedologic conditions, some soils with specific characteristics perform better than others.

The soils S3 and S4, characterized by a loam and sandy-loam texture, not very calcareous, with a sub-alkaline and neutral pH, low electrical conductivity, a content of organic matter between 5.46 and 8.67 g kg$^{-1}$, and a content of active lime between 21.25 and 26.25 g kg$^{-1}$, are favorable for the growth, stigma, and daughter corm production.

To obtain a high quality of the spice, it is not only the control of postharvest treatments such as drying and storage methods that are important, but also the choice of the soil. According to ISO

references, although all spice samples belonged to the first qualitative category, S1, S3 and S2 soils recorded the highest value for coloring power. Fewer variations of bittering and aromatic powers among soil type were found, even if the highest values of both powers were reached by S3 soil.

Soil type also influenced the formation of flower tepal and stamen (by-products), which can be further valorized and used in other sectors, increasing farming incomes and contributing to a sustainable cultivation of saffron.

Considering the importance of nutrient availability in saffron cultivation, further researches are recommended to assess how saffron yield, quality, and daughter corms production are influenced by other soil chemical properties, such as macronutrients content, mainly available phosphorus and potassium and total nitrogen in the soil [88].

It can be concluded that the assessment of soil conditions is particularly important to obtain the best saffron performance in terms of stigma and daughter corms yield as well as spice qualitative traits and by-products valorization. Moreover, the saffron corm growing system in pots with the use of selected soil can be considered as a suitable and alternative cultivation method to obtain a spice with high qualitative traits. This production type, appropriate for small farms based mainly on the work of family members, can guarantee the protection of the territory, avoiding the use of agricultural machinery for planting, fertilization, and weeding, and contribute to the vitality of rural areas in which the open-field cultivation is not possible or limited.

**Author Contributions:** Conceptualization, methodology and data curation, L.C., D.C., L.S., N.C. and V.C.; writing—Original draft preparation, L.C., D.C. and V.C.; writing—Review and editing, L.C., D.C., L.S., N.C. and V.C.; supervision, V.C. and M.P. All authors have read and agreed to the published version of the manuscript.

**Funding:** This research received no external funding.

**Acknowledgments:** The authors are grateful to Rocco Pirrone, Maria Montesano and Lorenzo Montinaro for their technical assistance.

**Conflicts of Interest:** The authors declare no conflict of interest.

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
