# Peer review of "The Influence of Soil Physical and Chemical Properties on Saffron (Crocus sativus L.) Growth, Yield and Quality"

_agronomy, doi:10.3390/agronomy10081154_

Round 1

Reviewer 1 Report

Agronomy-882419 : The influence of soil physical and chemical 2 properties on saffron (Crocus sativus L.) growth, yield and quality” by Cardone et al.

Cardone et al. present a generally well-written manuscript dealing with a topic that could be of interest to the Agronomy’s readers. The research objectives are clear, the methods and the experimental design are appropriate and articulated as well as the statistical analyses.

The results increase information to the body of knowledge about Saffron cultivation because it covers interesting applied aspects regarding growth, yield and quality of this economically important crop culture.

However, it seems to me that the biological components of soils are totally neglected in this MS.

For this reason, I would suggest to implement both introduction and discussion with some references regarding the different microbial components that might have impact on Saffron growth if associated with different soil types.

Soil is alive!

Specific Issue

What's about soil data regarding saffron production sites of Valle d'Aosta region?  Please, if possible, check them and add these references.

Line 142 This line is not legible

I would like to have a comment on the applicability of saffron cultivation out of their Italian common area of production using selected soil and pots system (as described) as an effective alternative cultivation strategy for the production of high quality saffron at familiar farming level.

Reviewer 2 Report

In the present manuscript (Agronomy-882419), the authors studied the effect of soil texture and chemical properties on Crocus sativus L. growth, yield, and saffron quality.

The quality of this work is very good. The present paper could be accepted in the current form, but minor revision is required beforehand. My comments and suggestions for author are outlined below.

  • Is it possible to add more recent references? Specially to replace references nº: 10, 11, 12, 13, 15, 18, 22, 63 and 71.
  • In line 142, there are some overlapping words at the beginning of the sentence.
  • In the current draft of the ISO 3632, the nomenclature named as “E” is replaced by “A”.
  • 1 cm” should be included as indicated in the Equation 1 (E1%, 1cm) every time the ISO parameter (E or A) are mentioned.
  • In lines 248-249, reference of this information (1 kg of dry stigmas, 256,410-140,845 flowers and 9.3-6.7 kg of by-products were needed) should be added.
  • In line 421, the parameters of ISO 3632:2011 are obtained by UV-vis spectrophotometry analyses, therefore it is not possible to refer to these results as “content”. It would be necessary to use other more specific techniques such as HPLC-DAD, NMR, GC, among others, in order to obtain the concentration (content) of the secondary metabolites of saffron. So, in this case, the results of the ISO 3632:2011 parameters should be expressed as values of E1%, 1cm440 nm (coloring strength),  E1%, 1cm330 nm, and  E1%, 1cm257 nm as the authors have written throughout the manuscript. This sentence should be re-written (line 142).

    I have attached a PDF file to better show what I mean in terms of how to write the parameter E (E1%, 1cm), because I could not insert formulas here.

Reviewer 3 Report

The main objective of this reseach is to relate soil properties of 7 soils (3 natural and 4 prepared) to saffron qulaity. For example, soil S3 is prepared from 33% S7  (sandy) added to natural S1 soil (clayey) ( I guess that is the way you did, it is not well elaborated in the text) and similarly soil S4 is preapred by adding soil S7 to soil S2 ( clay loam and not medium loam as you have written in the text).  Your conclusion is that soil chemical and physical properties of soil S3 and partly S4 is well related to saffron qulaity, yield and growth.  But in my opinion your data (although extensive and  infromational) is not supporting your conclusion.  There was no significant difference in yield, quality or their components of saffron grown insoils S1 to S4 (even S5 ) when they compared . Safarnal is the only quality component that significatly related to soil S3 because of more active lime content which needs more elaboration. That is the main strength point of this research.

An example,

Page 14/ Lines 349-: "Although S1 showed the highest values of flower and leaf traits, especially due to the its greatercontent of organic matter, it did not show the best results in terms of daughter corm production. We consider that the soil heavy texture could reduce the root growth."

Soil root growth is function of aeration, water avaiability and nutrients.  Soil S1 according to table 1 has lower bulk density,  more avaiable water and nutrient.  Root growth in soil S1 compared to soil S3 probably is not limited.

Soils S1,S2, and S7 calssification is required.

Table 7, there are two daughter corm number values? what is the difference?

Round 2

Reviewer 3 Report

my suggestions has been incorporated